# Cultural Differences in Design-Based Product Evaluation: The Role of Holistic and Analytic Thinking

Li Liu [1,*] and Ulrich Orth [2]

1   School of Economics and Management, China University of Petroleum-Beijing, Beijing 102249, China
2   A&F Marketing—Consumer Psychology, Christian-Albrechts-University, 24098 Kiel, Germany; uorth@ae.uni-kiel.de
*   Correspondence: liuli@cup.edu.cn

**Abstract:** Product evaluation research has a long tradition of examining how consumers evaluate a product from the product itself in an attempt to understand why certain products are better perceived or worse perceived. Usually consumers recall the memory of high evaluation products in their minds when they make buying decisions. Better fitting perceptions would be more favorable than poor fitting ones. Our findings indicate that culture is an important reason that influences consumers' responses to design-based product evaluations. Westerners evaluate products differently than Easterners due to cross-cultural differences in styles of thinking. Two cultures of people have differences in design-based product evaluation. In most cases, Easterners have more favorable evaluations of a new packaging product than Westerners.

**Keywords:** culture; product packaging; design elements; style of thinking





"Brain equals hardware, inferential rules and data processing procedures equal the universal software, and output equals belief and behavior, which can, of course, be radically different given the different inputs possible for different individuals and groups."

——Nisbett et al., 2001, p. 291

## 1. Introduction

Overseas marketing has always been complicated, because it is associated with foreign cultures. Understanding cultural diversities and nuances, and understanding your potential consumers, are vital for success in opening new foreign markets. In a mature foreign market, consumers have already had the ideal impression of a product; for example, gold foil contains excellent chocolates. Can product packaging that is popular in the home country be favored by foreign consumers? This is a very real problem; especially in the post-epidemic era, companies hope to expand overseas markets as soon as possible.

Marketing researchers have shown interest in product evaluation from visual stimuli for a long time. Key among them is the degree to which a new packaging design fits with the packaging of a known high-quality product [1–3]. Fit can be judged in a variety of ways, including whether the new packaging product has similar appearance attributes with existing high-quality products, such as material, size, etc., and whether the new packaging product expresses the same product information as the existing high-quality brand. Higher perceptions of packaging design fit result in more favorable product evaluations [4].

Little attention has been focused on the issue of whether these findings apply to consumers around the globe. Inspired by the research gap, the current research tries to answer the question whether cultural differences can influence product evaluation from visual stimuli, through addressing the following research questions: (1) What are the reasons which make cross cultural consumers have different evaluations of a new packaging product? (2) How to make a new packaging product more effective in cross-cultural marketing?

Unlike previous studies, which mainly applied survey research and case studies, we applied experiments to detect causal effect in cultural differences in design-based product evaluation. Through two experimental studies, we set up a series of new product packaging to test the evaluations of product qualities by consumers from different cultures. We contributed to the psychological mechanism that cultural influence has on consumer judgment by articulating the mediating role of cognitive thinking.

In the following sections, we first discuss the theoretical background of thinking style, product evaluation and propose research hypotheses. Next, we describe our research design and report the results of the two studies. Finally, we conclude with theoretical contributions and managerial implications, as well as limitations and future research.

## 2. Literature Review and Hypotheses Development

### 2.1. Cultures and Holistic/Analytic Thinking Style

Anthropological and psychological studies of general cognitive processes suggest that thinking styles are connected to culture [5,6]. Cultural differences in cognitive processes are tied to cultural differences in basic assumptions about the nature of the world [4]. Scholars in a number of disciplines have maintained that Asians and Westerners differed greatly in their methods of reasoning [7,8]. For Easterners, holistic reasoning can be summarized as orientation to the context or field as a whole, including attention to the relationships between a focal object and the field, a preference for explaining events on the basis of such relationships, and an approach that relies on experience-based knowledge rather than abstract logic and the dialectical. For Westerners, analytic reasoning can be summarized as a detachment of the object from its context, a tendency to focus on the attributes of the object in order to assign it to categories, a preference for using rules about categories to explain and predict an object's behavior, and inferences that rest in part on the decontextualization of structure from content, use of formal logic, and avoidance of contradiction [9–14].

It is generally believed that Asian culture rooted from China, as an example of how and why such a culture developed. The Chinese fostered a sense of collective agency. The individual was part of a close-knit group, according to Confucianism, the role fulfillment between emperor and subject, parent and child, older brother and younger brother was important [15]. Hence, individual rights were construed as one's "share" of the rights of the community as a whole [11]. Usually cultures rooted from Greece collectively labeled "Western cultures" are more analytic and independent. According to Hamilton [16], the Athenians were a union of individuals free to develop their own powers and live in their own ways. This location of power in the individual seems to be intimately related to the political organization (independent city-states) and the tradition of debate among the Greeks [17]. Western cultures were less concerned with context and social situations and tended to focus their attention more on individual objects and apply logic to what they see. After long period of time, people living in the East were more holistic, and people in the West were more analytic. Recent studies provided evidence that people in modern Eastern and Western cultures have inherited these ancient ways of thinking [13].

Analytic and holistic thinking theories have been used in practical marketing research. Researchers found that consumers from Eastern cultures, characterized by holistic thinking [18,19], perceive higher brand extension fit and evaluated brand extensions more favorably than consumers from Western cultures, characterized by analytic thinking [20]. These methods connect consumers' psychological thinking with final decision behavior. Therefore, our research is motivated by the assumption that Asians primarily focus on the field and on relationships, whereas Westerners primarily focus on objects and tend to detach objects from the field [21–23].

### 2.2. Packaging Design and Quality Evaluation

It is widely accepted that the packaging has an essential role in influencing consumer purchase choices and intentions in the process of purchase [24,25]. The packaging is the symbol that communicates favorable or unfavorable implied meaning about the product.

The past findings focused on the formation of the consideration set [26], product recall [27], brand evaluation [28].

In psychoanalytic theory, the procedure of information processing for consumers perceiving product packaging is made up of five sections [29,30].

Firstly, product packaging is exposed and noticed, consumers recognize and categorize some visual elements or their combinations. Next, they use the features of some stimuli according to their own subjective experiences in the past and cause meaningful information stimuli. The concept of packaging design is inherently multidimensional, incorporating multiple elements such as texts, shapes, graphic designs, logos, sizes, colors, illustrations, materials, textures and so on [31–33]. Furthermore, memory will affect received information and the way interprets it; memory is the result of learning. The working memory system is used to hold information actively in the mind and to manipulate that information to perform a cognitive task [34–36]. Meanwhile, the information which has been received will create memory [37]. The external condition is an important condition when doing design-based related research [38,39]. Optimal conditioning is the creation of a strong association between the conditioned stimulus and the unconditioned stimulus [40]; these two kinds of stimulation are usually contained in associative learning: implicit learning and explicit learning. Implicit learning is a primitive process of apprehending structure by attending to frequency cues, while implicit learning is without awareness, using rules. Our research is carried out in the two conditions: the implicit condition and explicit condition. After consumers accept messages and digest them into impressions, they can be used to interpret information for purchase or for decision-making [41,42]. A response model of consumers to product packaging is proposed in Figure 1.

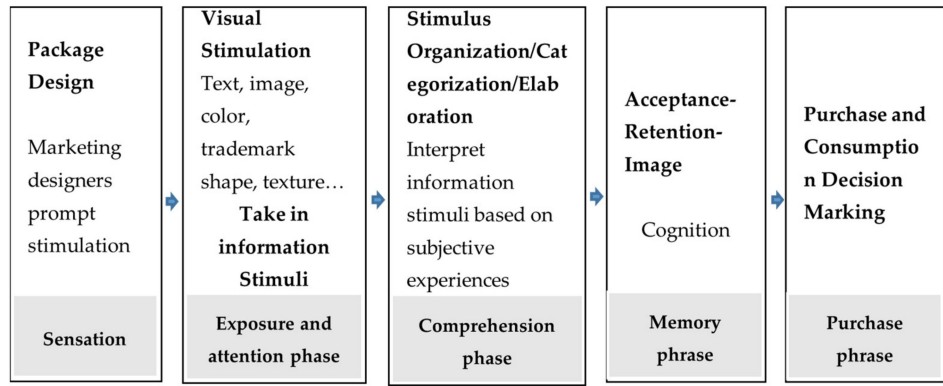

**Figure 1.** The procedure of perceiving product packaging.

### 2.3. Hypothesis and Research Model

According to the above theoretical description, when facing a new packaging product, higher perceptions of packaging design fit result in more favorable product evaluations by comparing with the packaging of known high-quality products. Consumers with holistic thinking find it easier to perceive higher fit of packaging design, while consumers with analytical thinking find it easier to perceive lower fit of packaging design. Westerners are more analytic people; Easterners are more holistic people.

In addition, the external condition is an important part of design-related research; our research is conducted under the implicit condition and explicit condition. In the real marketing environment, consumers exactly have these two buying conditions.

Above all, the following hypotheses are developed.

**H1.** *Consumers from Eastern cultures evaluate the product quality from packaging differently than consumers from Western cultures in the implicit condition.*

**H1a.** *From the perspective of whole packaging, consumers from Eastern cultures evaluate the product quality as higher from packaging than consumers from Western cultures in the implicit condition.*

**H1b.** *From the perspective of design elements, Western consumers perceive higher degrees of design elements than Eastern consumers in the implicit condition.*

**H1c.** *From the perspective of change levels, Western consumers perceive higher degrees of changes than Eastern consumers in the implicit condition.*

H1 hypothesizes that consumers from Eastern and Western countries have significant differences in their evaluation of a product from the product packaging due to the styles of thinking in the implicit condition. In the implicit condition, consumers are unable to see the known high-quality product packaging to help them to perceive the new product packaging during evaluation. H1 is tested in the perspective of the whole package, the perspective of design elements and the perspective of change levels.

**H2.** *Consumers from Eastern cultures evaluate the product quality from packaging differently than consumers from Western cultures in the explicit condition.*

**H2a.** *From the perspective of whole packaging, consumers from Eastern cultures evaluate the product quality as higher from packaging than consumers from Western cultures in the explicit condition.*

**H2b.** *From the perspective of design elements, Western consumers perceive higher degrees of design elements than Eastern consumers in the explicit condition.*

**H2c.** *From the perspective of change levels, Western consumers perceive higher degrees of changes than Eastern consumers in the explicit condition.*

H2 hypothesizes that consumers from Eastern and Western countries have significant differences in their evaluation due to the thinking styles in the explicit condition. In the explicit condition, consumers are able to see the known high-quality product packaging to help them to perceive the new product packaging during evaluation. H2 is tested in the three perspectives as study 1.

The research model is presented in Figure 2. It shows that visual cues will affect product evaluations by associative learning; people make a judgment through what they see. In this research, all the visual cues are from package design, called design-based visual cues. In the processing of judging from visual cues, holistic and analytical thinking work. Holistic or analytic thinking is considered as a mediator in the process of visual cues and product evaluation. The main purpose of this research is to test that culture works in design-based impression formation with holistic and analytic thinking under controlled conditions— the implicit and explicit condition, respectively.

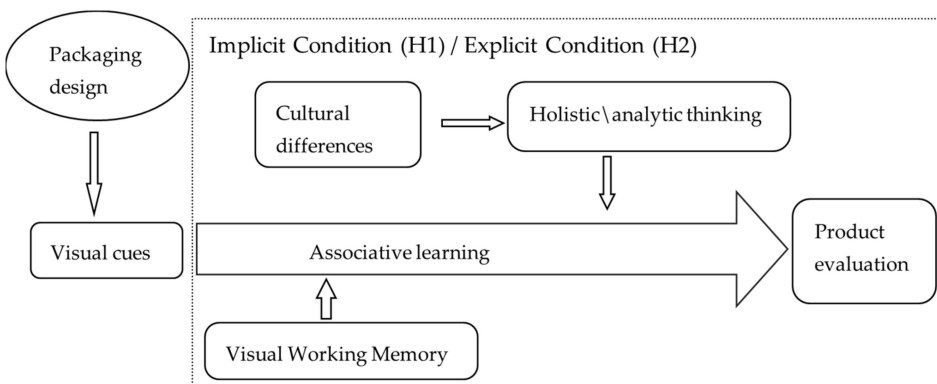

**Figure 2.** The research model.

## 3. Methodology and Results

### 3.1. Experimental Design

In the process of experiment design, we determined wine bottles as research objects. The wine bottle is a bottle used for holding wine, generally made of glass, because it is always glass-made, and the design elements can be easily found on it, such as shape and

size. This advantage brings a lot of convenience to showing participants certain design elements which we focus on. Moreover, wine is a basic and familiar food product in Western and Eastern countries.

What design elements to choose? The visual elements are divided into two parts: (a). packaging graphics; and (b) packaging sizes and shapes. Specifically, packaging graphics are further divided into packing layout, image, color, and typography [43]. In pretesting, there were 12 items of elements listed, some of them were not visual elements but informational elements. We invited 12 German students and 12 Asian students to identify visual related items. As we expected, eight visual design elements were left at last. Color is important, but it is a quite complex matter in people's perception, consumers seem to have personal and cultural preferences for some colors over others [44], and the size of wine bottle are always the same, 750 mL. We eliminated these two elements. At last, six selected design elements were shape of bottle, logo, font of brand name, shape of cap, image on bottle and its position.

How to find a standard wine bottle with six selected design elements? Because this research was a comparison research, a standard wine bottle needed to be found. This bottle should be formed by the six design elements above. We collected 44 wine bottles of varied prices in Kiel Germany, and 19 wine bottles from wine shops in Beijing China. With the help of design experts we extracted the six elements from 63 wine bottles, determined the standard elements, respectively. The principle of selection of standard elements is to select the most common, basic and least product information elements and then combined the six elements together to define the standard wine bottle. The new standard bottle is showed in Figure 3.

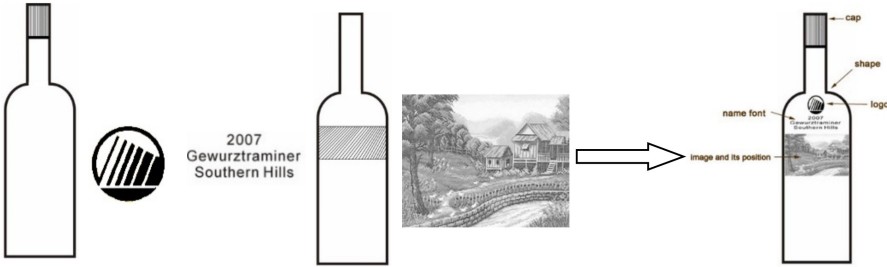

**Figure 3.** The formation of the standard bottle.

How to determine the change level of each element? Each element has two levels of variation based on the standard element, the small changed level and the big changed level. In the pretest, for example, the shape of bottle had five gradually changing patterns, and these changes were all based on the confirmed standard shape of bottle. A total of 13 German students and 12 Asian students were invited to identify the small changed element and the big changed element separately. According to the pretest, we determined the change level of each element finally. The small changed and big changed elements are shown in Figure 4.

How to know who are holistic or analytic persons? The Embedded Figure Test (EFT) is designed to measure disembedding, a restructuring skill, which results from the use of styles of thinking and a measure of both cognitive method and analytical ability, and involves detecting simple figures embedded in larger, more complicated figures [4]. There were 24 figures in the EFT. All participants were asked to do the EFT and given 10 min to complete the test.

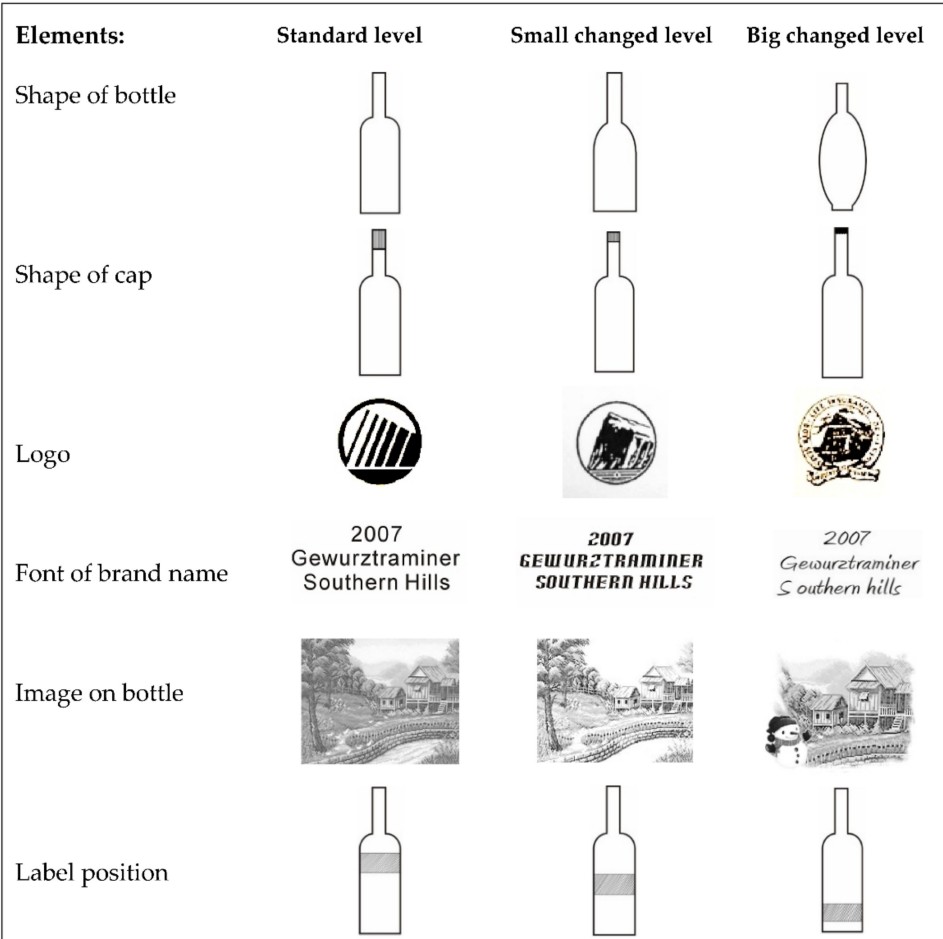

**Figure 4.** Three change levels of six design elements.

*3.2. Study 1*

3.2.1. Process

In study 1, we selected a level from each element to form a new bottle. The new formed bottles are the objects which were showed to participants. Participants are all under the implicit condition. This means participants could not see the original standard bottle when evaluating quality. Instead of testing all possible combinations, we adopted the method of Orthogonal Experimental Design to find the target new bottles using orthogonal arrays (OA) to organize the parameters affecting the process and the levels at which it should be varied. It can help to collect the necessary data and determine which factors most affect experimental results with a minimum amount of experimentation, thus saving time and resources [45]. A total of 18 new bottles were selected.

A total of 62 students including 36 females and 26 males, enrolled in a cross-cultural marketing course at the University of Kiel, were recruited for the Germany sample; a total of 64 participants including 34 females and 30 males were recruited for the Chinese sample. None of the participants had a cross-cultural background. Their ages (mean of Chinese, $M_{CN}$ = 24.7; mean of German, $M_{DE}$ = 23.2) were less than 25. None of the participants had a cross-cultural background. They are existing or potential consumers in the wine market and the target audience in our research.

In the first phase, called the "learning" phase, people had to learn the standard bottle which was formed from six standard elements (shape, logo, cap, font of brand name, image and its position). In this period, participants saw the standard bottle as long as they wanted, then began to the learn six elements separately. In the process of element learning, two wrong figures also appeared to disturb and enhance participants' memory of the standard elements. They needed to choose the standard element from 3 figures (standard, small

changed and big changed elements). In the case of participants choosing the wrong figure, they were not allowed to go on to the next question. Participants had to choose again until they found the right answer.

In the second phase, participants were told that the standard bottle contained the greatest wine, marked 101 scores and learned that the quality of wine only depended on the packaging (bottle). The more similar it is to the standard bottle, the higher quality of wine in the bottle and vice versa. Participants were asked the question: "how do you think about the quality of wine in this bottle?" and gave their evaluation scores from 1 to 101 scale (1 = extremely bad, and 101 = extremely good) in 10 s. They needed to evaluate 18 bottles of wine which was selected by the orthogonal design method. The 18 bottles were randomly shown to participants.

Finally, participants did the EFT to test their cognitive styles.

### 3.2.2. Results

A one-way ANOVA (Analysis of Variance) with culture as the independent variable indicated that, as anticipated, German participants were significantly more oriented toward analytical processing than Chinese participants ($M_{CN}$ = 1.375; $M_{DE}$ = 2.516; $p < 0.00$) from EFT.

To test cultural differences in quality evaluation, first, we show the results of means of the whole packaging quality evaluation in the implicit condition in Figure 5; it indicates that Chinese participants gave higher scores of quality evaluation than German participants. In addition, a separate ANOVA was performed for each new combined bottle, with culture (Western, Eastern) as an independent variable and quality scores as the dependent variable. One-way ANOVA was used for 18 groups of data. As expected, a significant main effect of culture emerged for each analysis.

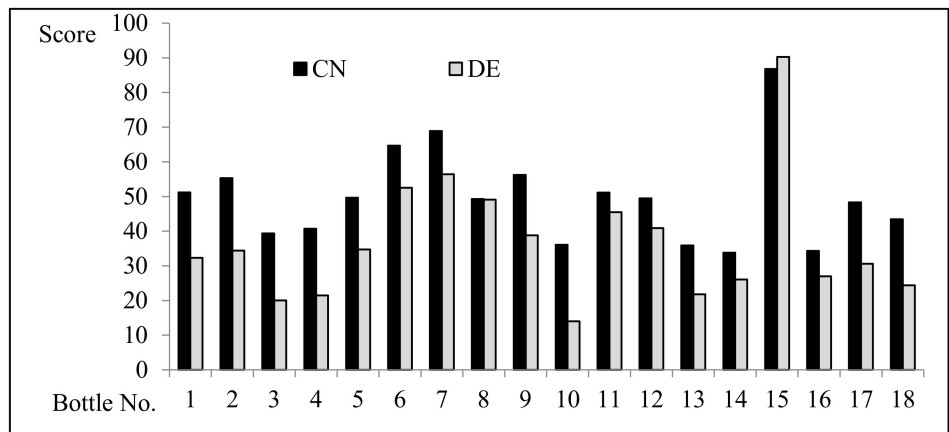

**Figure 5.** Means of the whole packaging quality evaluation in the implicit condition CN is the abbreviation for the Chinese sample, DE is the abbreviation for the German sample. The evaluated bottles are marked as No.1 bottle, No.2 bottle, No.3 bottle . . . .

Next, a mediation analysis was conducted to test whether style of thinking (holistic and analytic thinking) was a mediator of cultural differences in perceptions of quality evaluation. Three regression analyses were performed in the mediation analysis [4]. The results are shown in Table 1.

**Table 1.** Results of mediation analyses in study 1.

| Condition | Regression Equations |
|---|---|
| 1 | Culture (0.417 ***) influences the type of thinking |
| 2 | Culture (−0.233 ***) influences the type of perceived quality |
| 3 | The type of thinking influences (−0.206 **) the quality evaluation and decreases the influence of culture (−0.063 **) on the quality evaluation<br>Sobel's Z = 2.80, *p* = 0.005 ** |

** $p < 0.01$; *** $p < 0.001$.

The result shows that the type of thinking is the mediator of the influence of culture on quality evaluation. The type of thinking significantly influences quality scores in Equation (3), while culture (−0.063) also influences quality scores. Culture (−0.063) in Equation (3) has less influence on the quality evaluation than culture (−0.233) in Equation (2). It is confirmed that the type of thinking is the mediator between culture and quality evaluation. H1a is supported.

As mentioned in the previous experimental process, the advantage of OA is to help us estimate the contribution of individual influencing factors in experiments. To test the influence of each design element on quality evaluation, we conducted ANOVAs to analyze the scores of product evaluation in the orthogonal array experiment. The results of the influence of six design elements on quality evaluation are shown in Table 2.

**Table 2.** Results of six elements affecting quality evaluation in the implicit condition.

| Source | | Chinese Sample | | | | German Sample | | | |
|---|---|---|---|---|---|---|---|---|---|
| | Changed Level | M | SD | F | Sig. | M | SD | F | Sig. |
| Image | Standard | 53.40 | 2.56 | | | 41.80 | 5.13 | | |
| | Small changed | 48.67 | 5.23 | 7.5 | 0.00 | 37.53 | 6.12 | 26.63 | 0.00 |
| | Big changed | 46.88 | 7.12 | | | 30.66 | 6.09 | | |
| Shape of bottle | Standard | 63.40 | 3.22 | | | 50.48 | 3.23 | | |
| | Small changed | 48.86 | 2.75 | 118.12 | 0.00 | 37.80 | 4.89 | 175.36 | 0.00 |
| | Big changed | 36.68 | 8.01 | | | 21.71 | 4.72 | | |
| Logo | Standard | 54.32 | 8.06 | | | 44.44 | 6.44 | | |
| | Small changed | 47.78 | 8.96 | 10.98 | 0.00 | 39.91 | 6.11 | 39.29 | 0.00 |
| | Big changed | 46.84 | 5.22 | | | 31.64 | 8.96 | | |
| Label position | Standard | 50.59 | 5.23 | | | 43.10 | 4.56 | | |
| | Small changed | 49.33 | 9.74 | 0.46 | 0.64 | 35.91 | 7.96 | 31.29 | 0.00 |
| | Big changed | 49.02 | 7.33 | | | 30.98 | 8.11 | | |
| Shape of cap | Standard | 55.78 | 2.03 | | | 45.32 | 1.33 | | |
| | Small changed | 48.45 | 4.32 | 20.93 | 0.00 | 35.53 | 5.14 | 56.02 | 0.00 |
| | Big changed | 44.71 | 4.15 | | | 29.14 | 4.31 | | |
| Font of brand | Standard | 53.40 | 4.23 | | | 44.24 | 4.56 | | |
| | Small changed | 47.25 | 7.79 | 7.17 | 0.00 | 29.94 | 6.34 | 43.27 | 0.00 |
| | Big changed | 48.29 | 7.01 | | | 35.85 | 5.69 | | |

M is the abbreviation for mean value, F is the abbreviation for F value and Sig is the abbreviation for Significance.

From Table 2, all six elements have an impact on the quality evaluation in the German sample, but five of the six elements significantly influenced the quality evaluations in the Chinese sample. The label position was not significant in Chinese sample, it could be evidence to prove that Eastern consumers perceive lower degrees of design elements than Western consumers in the implicit condition. We also found that Chinese participants gave higher scores of all levels of design elements than German participants. The Figure 6 show the means of the six design elements in the implicit condition, respectively.

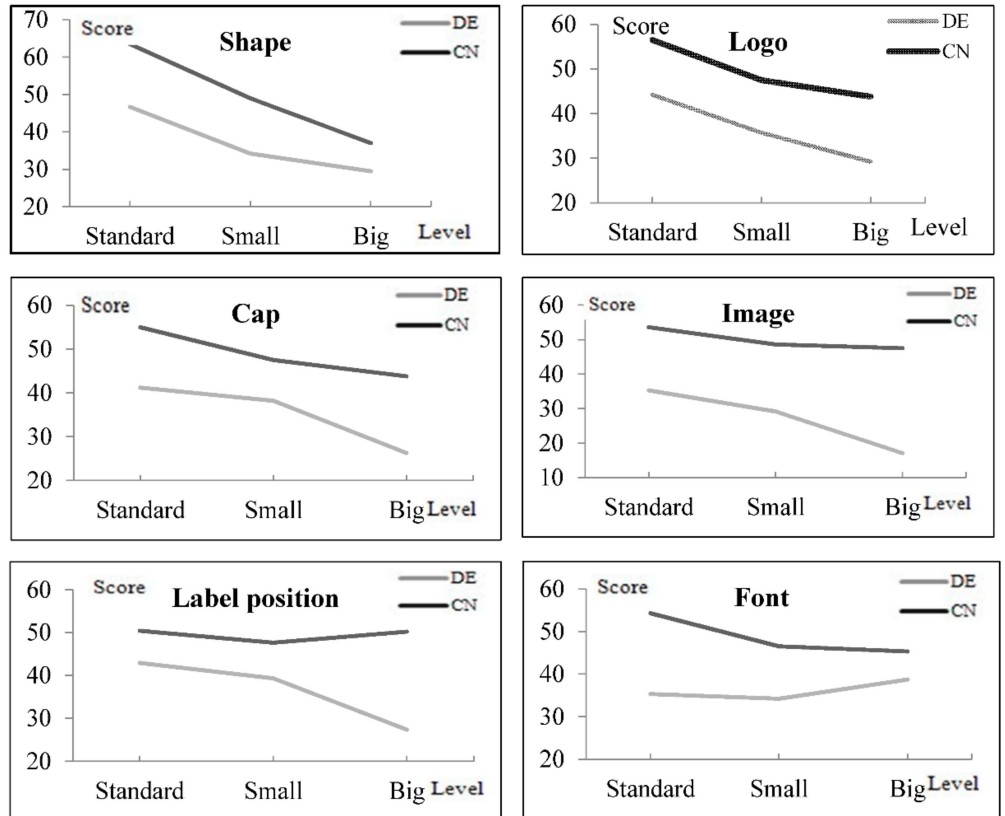

**Figure 6.** Means of the six design elements in the implicit condition CN is the abbreviation for the Chinese sample, DE is the abbreviation for the German sample.

On the one hand, Chinese participants gave higher scores in all levels and elements than German participants and Chinese gave higher scores of every element level. On the other hand, Chinese and German participants had different sensitivities to the changes of elements. In most elements, means from the German sample dropped sharply. For example, the design element of the cap from the small changed level to big changed level. However, Chinese were less sensitive to the changes of elements, for example, label position. Above all, H1b was certified.

In order to test H1c, we divided the 18 bottles into five categories according to the scores of previous evaluations. The five categories show extremely high similarity (81–101), high similarity (61–80), moderate similarity (41–60), low similarity (21–40) and extremely low similarity (0–20). We selected the bottle with the smallest standard deviation in each category. A 2 (culture) × 5 (similarity level) ANOVA was performed, with wine familiarity as a covariate. As expected, a significant main effect of culture ($F(1, 125) = 89.3$, $p < 0.00$) and similarity levels ($F(4,125) = 63.4$, $p < 0.00$) emerged. In the extreme low similarity level, $M_{CN} = 35.9$, SD = 8.36, $M_{DE} = 14.5$, SD = 9.43, $p < 0.00$; in the low similarity level, $M_{CN} = 49.2$, SD = 6.11, $M_{DE} = 40.7$, SD = 7.98, $p < 0.05$; in the moderate similarity level, $M_{CN} = 40.9$, SD = 5.31, $M_{DE} = 22.5$, SD = 3.21, $p < 0.00$; in the high similarity level, $M_{CN} = 45.9$, SD = 9.98, $M_{DE} = 20.8$, SD =8.98, $p < 0.00$; in the extreme high similarity level, $M_{CN} = 75.9$, SD = 5.63, $M_{DE} = 52.5$, SD = 7.01, $p < 0.01$. There was no interactive effect among levels and cultures. Levels * cultures were not significant. Means of the different levels of change show that Chinese perceived higher scores of quality evaluation than Germans. From the perspective of change levels, Western consumers perceive higher degrees of changes than Eastern consumers in the implicit condition, H1c was supported.

### 3.2.3. Discussion

In study 1, the results of the holistic and analytic thinking test support Easterners and Westerners having distinct styles of thinking. When comparing the scores of Germans and

the Chinese participants' evaluation, consumers from Eastern culture evaluated quality more favorably than consumers from Western culture from the whole packaging change perspective. In most cases, six design elements influence the judgement of product quality, Chinese (Easterners) perceived higher scores than Germans (Westerners) in three levels of each element. When comparing the sensitivity of Germans and Chinese towards changes, we found that participants altered their evaluations when the new package changed, but Western consumers perceive higher degrees of changes than Eastern consumers in the implicit condition in total. Therefore, Consumers from Eastern cultures evaluate the product quality from packaging differently than consumers from Western cultures in the implicit condition. Consumers from Eastern cultures evaluated quality more favorably than consumers from Western cultures in the implicit condition.

*3.3. Study 2*

3.3.1. Process

Study 2 was under a new condition—the explicit condition. In the explicit condition, participants had a process of explicit learning. In this process, rule discovery is the inductive way. In a real market this is happening every minute, consumers can see the original high quality of products when they make a buying decision. In our experiment, participants can receive tips from the original standard bottle when evaluating new bottles. They can compare the new bottle with the standard bottle directly. A total of 62 persons including 33 females and 29 males, were recruited for the Germany sample from students enrolled in a marketing research course at the University of Kiel. A total of 72 subjects including 36 females and 36 males, all local students in China, were recruited for the Chinese sample. Their ages ($M_{CN}$ = 25.9, $M_{DE}$ = 24.8) were around 25, they are existing or potential consumers in wine market.

In the first phase, participants had the same experimental process as study 1, learning the standard elements and the standard bottle. In the second phase, they were told that the standard bottle contained the greatest wine, marked 101 scores and the quality of wine only depended on the packaging (bottle). The more similar it is to the standard bottle, the higher quality of wine in the bottle and vice versa. A total of 18 combined bottles that were selected by the orthogonal design method were randomly displayed to the participants, and the standard bottle appeared at the same time. Participants were asked the question: "how do you think about the quality of wine in this bottle?" and gave their evaluation scores from 1 to 101 (1 = extremely bad, and 101 = extremely good) in 10 s. They needed to evaluate all 18 bottles of wine. At last, participants did the EFT (Embedded Figures Test) to test their cognitive styles.

3.3.2. Results

The results of EFT indicated that German participants were significantly more oriented towards analytical processing than Chinese participants ($M_{CN}$ = 0.97; $M_{DE}$ = 1.59; $p$ <0.01).

As in study 1, separate one-way ANOVA was performed for each combined bottle, with culture (Western, Eastern) as an independent variable and quality scores as the dependent variable. One-way ANOVA was used for 18 groups of data. As expected, a significant main effect of culture emerged from each analysis. Chinese participants perceived higher scores of quality evaluation than German participants in the explicit condition, shown in Figure 7. For the whole of the packaging, consumers from Eastern cultures evaluated quality more favorably than consumers from Western cultures in the explicit condition.

A mediation analysis was also conducted to test whether styles of thinking (holistic and analytic thinking) were a mediator of cultural differences in perceptions of quality evaluation. Three regression analyses were performed in the mediation analysis [4]. The results are shown in Table 3.

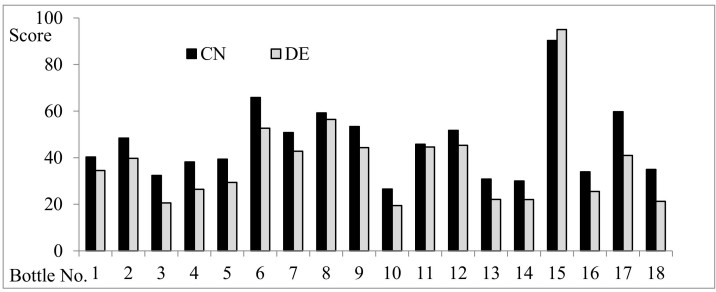

**Figure 7.** Means of the whole packaging quality evaluation in the explicit condition. CN is the abbreviation for the Chinese sample, DE is the abbreviation for the German sample. The evaluated bottles are marked as No.1 bottle, No.2 bottle, No.3 bottle . . . .

**Table 3.** Results of mediation analyses in study 2.

| Condition | Regression Equations |
|---|---|
| 1 | Culture (−0.210 **) influences the type of thinking |
| 2 | Culture (−0.141 ***) influences the type of perceived quality |
| 3 | The Type of thinking influences (0.143 **) quality evaluation and decrease the influence of culture (−0.013) on quality evaluation |

** $p < 0.01$; *** $p < 0.001$.

The results show that the type of thinking is the mediator of the influence of culture on quality evaluation in the explicit condition. First, the style of thinking significantly influences quality scores in the Equation (3), while culture (0.141) also influences quality score. Culture (−0.013) in Equation (3) has less influence on the quality evaluation than culture (0.141) in Equation (2). From above results, consumers from Eastern cultures evaluate quality more favorably than consumers from Western cultures in the explicit condition; H2a was confirmed.

To test the influence of each design element on quality evaluation, as in study 1, we conducted ANOVAs to analyze the scores of product evaluation. The results are shown in Table 4.

**Table 4.** Results of six elements affecting quality evaluation in the explicit condition.

| Source | Changed Level | Chinese Sample | | | | German Sample | | | |
|---|---|---|---|---|---|---|---|---|---|
| | | M | SD | F | Sig. | M | SD | F | Sig. |
| Image | Standard | 48.36 | 2.48 | | | 44.41 | 2.33 | | |
| | Small changed | 41.71 | 4.56 | 6.14 | 0.00 | 36.15 | 3.96 | 27.45 | 0.00 |
| | Big changed | 40.91 | 5.02 | | | 30.42 | 3.99 | | |
| Shape of bottle | Standard | 59.72 | 1.36 | | | 54.30 | 1.99 | | |
| | Small changed | 40.95 | 2.56 | 81.44 | 0.00 | 35.16 | 2.88 | 150.47 | 0.00 |
| | Big changed | 30.31 | 3.84 | | | 21.52 | 2.87 | | |
| Logo | Standard | 48.60. | 3.39 | | | 44.19 | 5.22 | | |
| | Small changed | 42.20 | 4.31 | 7.09 | 0.00 | 35.37 | 4.88 | 23.73 | 0.00 |
| | Big changed | 40.19 | 4.19 | | | 31.426 | 7.97 | | |
| Label position | Standard | 50.11 | 5.42 | | | 43.46 | 8.36 | | |
| | Small changed | 41.92 | 4.21 | 12.27 | 0.00 | 35.83 | 6.37 | 19.81 | 0.00 |
| | Big changed | 38.95 | 1.65 | | | 31.68 | 7.33 | | |
| Shape of cap | Standard | 49.85 | 2.12 | | | 43.72 | 7.66 | | |
| | Small changed | 43.03 | 2.31 | 12.76 | 0.00 | 36.68 | 7.14 | 24.01 | 0.00 |
| | Big changed | 38.11 | 5.02 | | | 30.58 | 8.56 | | |
| Font of brand | Standard | 46.89 | 1.89 | | | 43.33 | 2.3 | | |
| | Small changed | 42.41 | 5.12 | 2.93 | 0.05 | 33.36 | 4.30 | 16.81 | 0.00 |
| | Big changed | 41.68 | 6.33 | | | 34.29 | 4.13 | | |

In study 2, the six design elements all influenced the quality evaluation in the German sample and Chinese sample. Chinese participants gave higher scores of all levels of design elements than German participants. The Figure 8 show the means of the six design elements in the explicit condition, respectively.

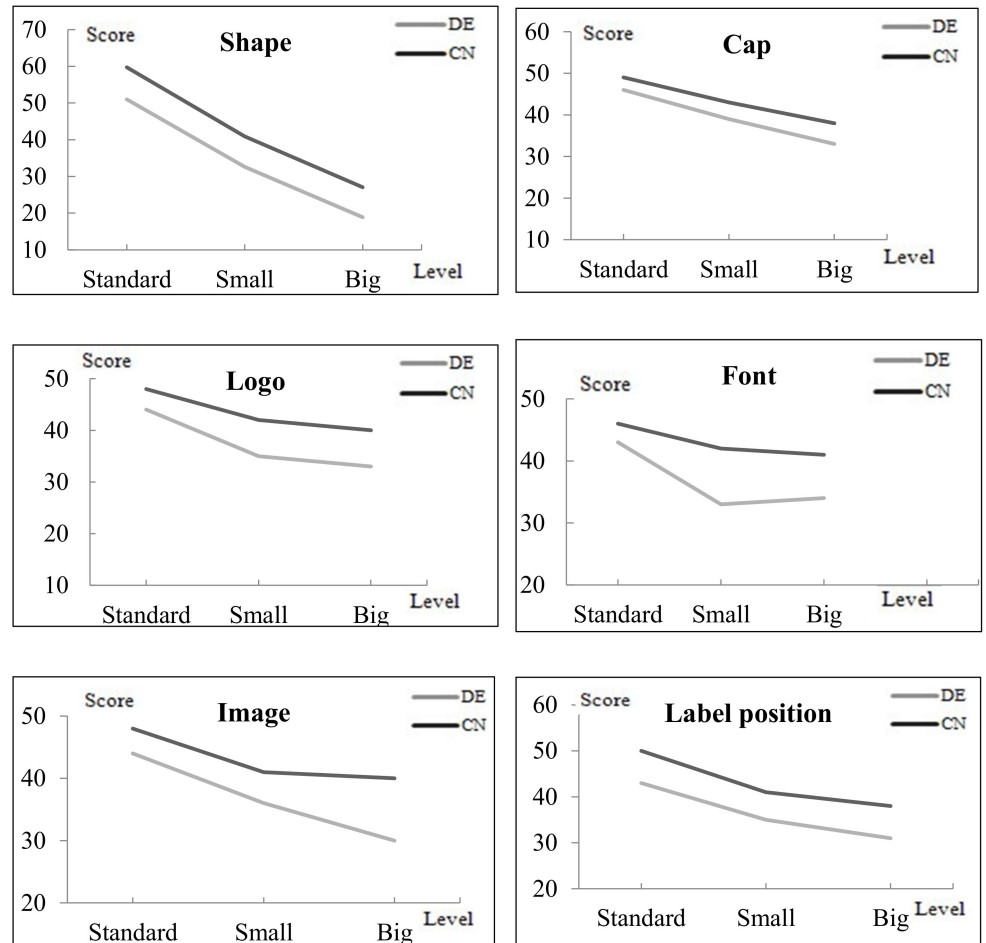

**Figure 8.** Means of six design elements in the explicit condition. CN is the abbreviation for the Chinese sample, DE is the abbreviation for the German sample.

Germans gave significantly lower scores than Chinese gave in the same changed level. It is worth mentioning that Chinese and German participants in the explicit condition became more sensitive to the changes of elements, because the gap of scores between the levels are bigger than that in the implicit condition. Above all, H2b was supported.

In order to test H2c, as in study 1, 18 bottles were divided into five categories. A 2 (culture) $\times$ 5 (similarity levels) ANCOVA was performed. As expected, a significant main effect of culture ($F_{(1, 153)} = 23.2$, $p < 0.00$) and similarity levels ($F_{(4, 153)} = 65.9$, $p < 0.00$) emerged. In the extreme low similarity level, $M_{CN} = 37.6$, SD = 9.11 $M_{DE} = 18.5$, SD = 10.32, $p < 0.00$; in the low similarity level, $M_{CN} = 39.2$, SD = 4.51, $M_{DE} = 30.7$, SD = 7.58, $p < 0.05$; in the moderate similarity level, $M_{CN} = 45.9$, SD = 5.42, $M_{DE} = 31.5$, SD = 8.33, $p < 0.00$; in the high similarity level, $M_{CN} = 49.9$, SD = 7.6, $M_{DE} = 40.8$, SD = 10.21, $p < 0.00$; in the extreme high similarity level, $M_{CN} = 73.9$, SD = 4.01, $M_{DE} = 60.4$, SD = 10.20, $p < 0.01$. The quality scores rose as the similarity increased from the extreme low similarity to the extreme similarity. The interaction culture * similarity level was not significant ($F_{(4, 153)} = 1.2$, $p = 0.31$). From the perspective of change levels, Western consumers perceive higher degrees of changes than Eastern consumers in the explicit condition. H2c was supported.

### 3.3.3. Discussion

From the results of study 2, it could be seen that consumers from Eastern cultures evaluate quality from packaging differently than consumers from Western cultures in the explicit condition. From the whole packaging perspective, consumers from Eastern cultures evaluate the product quality from packaging as higher than consumers from Western cultures in the explicit condition. All elements affected the quality evaluation under the explicit condition from the design elements perspective; the means of every changed level showed that Eastern consumers perceive higher degrees of elements changes than Western consumers. From the change levels perspective, Eastern consumers perceived higher degrees of packaging changes than Western consumers did. When comparing the sensitivity of Germans and Chinese towards changes, we found that the gap of scores from the standard to extremely low similarity was bigger than that of study 1.

### 3.4. Comparison of the Implicit and Explicit Conditions

### 3.4.1. Comparison of Whole Package Evaluation in Two Conditions

To test the implicit and explicit conditions and whether they influence the relationship between culture and quality evaluation, separate ANOVAs were performed in Germans and Chinese, with conditions (implicit, explicit) as an independent variable and quality scores as the dependent variable. In Germans sample, the evaluations of two bottles, bottle No.10 ($F = 5.08$, $p < 0.03$) and bottle No.17 ($F = 5.99$, $p < 0.02$), were significantly different in the implicit and explicit conditions among the 18 bottles. In the Chinese sample, the evaluation of five bottles, bottle No.1 ($F = 4.89$, $p < 0.03$), bottle No.5 ($F = 5.37$, $p < 0.02$), bottle No.7 ($F = 5.77$, $p < 0.01$), bottle No 10 ($F = 4.53$, $p < 0.04$) and bottle No.17 ($F = 6.35$, $p < 0.01$), were significantly different in the implicit and explicit conditions among the 18 bottles. It can be concluded that the influence of conditions on Germans' perception of quality evaluation is less than that of Chinese in our experiment.

### 3.4.2. Comparison of Sensitivity in Two Conditions

As known, there were five levels of bottles (extreme low similarity; low similarity; moderate similarity; high similarity and extreme high similarity). The score's gap was defined as $\Delta$Score. $\Delta$Score was a group of new data which represented the score gap between levels. $\Delta Score_1 = Score_{extreme\ low\ similarity} - Socre_{low\ similarity}$; $\Delta Score_2 = Score_{low\ similarity} - Socre_{moderate\ similarity}$; $\Delta Score_3 = Score_{moderate\ similarity} - Socre_{high\ similarity}$; $\Delta Score_4 = Score_{high\ similarity} - Socre_{extreme\ high\ similarity}$. $\Delta Score_{1ex}$ means $\Delta Score_1$ of the explicit condition; $\Delta Score_{1im}$ means $\Delta Score_1$ of the implicit condition, and so on. We can know from Table 5 that Germans had no significantly different sensitives towards levels of change in two conditions, but Chinese had significant different sensitivities from the extremely low similarity level to the low similar level in the implicit and explicit conditions.

**Table 5.** Comparison of the sensitivity in two conditions.

| Sensitivity | German Sample | | Chinese Sample | |
|---|---|---|---|---|
| | F | Sig. | F | Sig. |
| $\Delta_{Score1ex}$ vs. $\Delta_{Score1im}$ | 0.068 | 0.795 | 11.81 | 0.00 |
| $\Delta_{Score2ex}$ vs. $\Delta_{Score2im}$ | 0.013 | 0.910 | 2.53 | 0.11 |
| $\Delta_{Score3ex}$ vs. $\Delta_{Score3im}$ | 02.27 | 0.134 | 2.72 | 0.10 |
| $\Delta_{Score4ex}$ vs. $\Delta_{Score4im}$ | 0.014 | 0.904 | 0.31 | 0.58 |

## 4. Conclusions and Implications

### 4.1. Conclusions

Product evaluation research has a long tradition of referring to examining how consumers evaluate product from product packaging itself to understand why certain products have high or poor evaluation. When consumers make buying decisions, they always recall the memory of high evaluation products. The better fitting perception of new packages



would be more favorable evaluation than the poor fitting ones. This research explores how it works on cultures. The key finding indicates that culture is an important reason that influences consumer response to product evaluations. Westerners evaluate products differently than Easterners due to cross-cultural differences in styles of thinking. Two cultures of people have differences in design-based product evaluation. In most cases, Easterners provided more favorable evaluations of a new packaging design product than Westerners did.

### 4.2. Theoretical Contributions

For cross-cultural research, it adds to the growing body of research that supports that culture is dynamic. The research shows that culture operates by making certain forms of thinking more accessible than others. It reaches an important step that links the analytic–holistic thinking with physical property mapping–relational linking. In the meantime, it adds to the literature that culture has an important influence on consumer behavior-related issues. Based on our research, researchers could make more specific predictions regarding cultural differences in evaluation perception. Culture, thinking and consumer behavior collect together, and more research can be done in future.

### 4.3. Managerial Implications

Marketers always face a problem when they explore overseas markets, because they are not sure what rules they should follow to reduce the risk of extending their new product to a market that has had high value products. Packaging perception is the first perception of a new product. The perception of product evaluation can be varied. Consumers of analytic thinking have a higher perception for the change, whereas consumers of holistic thinking have a lower perception for the same change. When a new packaging product begins to sell in a new market, they need to investigate whether their potential consumers are more holistic or analytic. Proper marketing strategies can help marketers overcome the challenges from new packaging products. In addition, two consuming environments (the implicit condition and explicit condition) had the same trends of product perceptions, but the influence of conditions on Westerners' perception of product evaluation is less than that of Easterners' perceptions of product evaluation.

### 4.4. Limitations and Future Research

Although we deeply explored cultural differences in design-based product evaluation, making contributions theoretically and practically, there are still two limitations in this research that could be discussed in future studies. First, products can be divided into many categories; they have different values to consumers. In this research, we did not explore other categories of products which could be further explored. Second, cultural psychology is that cultural practices influence psychological processes, which in turn transform cultural practices [46]. Given these mutual influences, it is possible that the nature of consumer styles of thinking leads to cultural practices over long enough time. Deeper discussions from the perspective of cultural differences in perceiving products may help us better understand things happening in product evaluation from visual stimuli.

**Author Contributions:** Conceptualization, L.L. and U.O.; writing, L.L.; methodology, L.L. and U.O.; software, L.L.; investigation, L.L. and U.O.; supervision, U.O.; funding acquisition, L.L. All authors have read and agreed to the published version of the manuscript.

**Funding:** This research was funded by the Humanities and Social Sciences Youth Foundation of the Ministry of Education of China, grant number 19YJC840023 and Science Foundation of China University of Petroleum, Beijing, grant number 2462020YXZZ041.

**Institutional Review Board Statement:** Not applicable.

**Informed Consent Statement:** Not applicable.

**Data Availability Statement:** Not applicable.

**Acknowledgments:** The authors want to acknowledge all the participants taking part in the experiments.

**Conflicts of Interest:** The authors declare no conflict of interest.

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
