# Peer review of "Cultural Differences in Design-Based Product Evaluation: The Role of Holistic and Analytic Thinking"

_sustainability, doi:10.3390/su13052775_

Round 1

Reviewer 1 Report

The paper presents the study performed on two types of questionnaires carry out to students in both a Chinese and a German university. More specifically, both questionnaires asked to evaluate on a scale of 1 to 101 the quality of a series of products (bottles of wine on which some variables were controlled by the authors) based on the following statement "The more similar it is to the standard bottle, the higher quality of wine in the bottle and vice versa". One group of participants (both Chinese and German) were allowed to view the standard product during their responses (explicit condition) while another group was not allowed (implicit condition). The analysis of the results allowed the authors to confirm 6 hypotheses that emerged during the discussion of the state of the art.

I propose below the main criticalities of the paper

Evaluations are not about the perception of quality but the perception of the similarity of the product shown to a standard one. Therefore, any discussion related to preference and/or quality understood as the ability to meet expectations is superfluous.

The only difference between the first three hypotheses and the second three is the type of context (explicit or implicit) and, in both contexts, the hypotheses lead to the same conclusions. Authors should reformulate the hypotheses and/or present and discuss the differences that emerged between the two contexts.

Standard deviations are not reported in all graphs. I strongly recommend reporting the variability of the responses in addition to the averages to verify, even visually, the significance of the results obtained and discussed. Especially for testing hypotheses b and c, the slope of the graphs due to the averages alone is not enough to test the hypotheses.

The difference between explicit and explicit context should be better introduced in the introduction.

A minor limitation of the study is that Chinese students were taken as representative of those in the East and German students as representative of those in the West. However, other European countries such as France and Italy have a tradition of production and consumption of wine that is much more deeply rooted than that of Germany. It seems a bit far-fetched to assume that the responses of German students are representative of the responses that Italian and French students might have given.

Reviewer 2 Report

The authors are to be congratulated for addressing a topic that is so little studied at present and yet has important implications for the interpretation of packaging design. The authors are also commended for an elegant review of the literature.

In order to improve the authors' study the following changes are advised:

- The description of lines 189 to 193 requires a more extended explanation, as well as the use of pictures to help understand the methodology and its appropriateness.

- Some formatting changes are advised, such as Figure 3 being split into two pages, or the heading 3.2.2 Results (line 242) being on another sheet.

- In point 3.2 there is a lack of data on the target audience: age, gender, country of origin, relationship of people with this type of packaging, etc.

- In point 3.2 Are they the same students?

  • Line 182: 12 German and 12 Asian students were invited.
  • Line 197: 13 German and 12 Asian students were invited.

- Line 206: Is The Embedded Figure Test (EFT) referenced in citation 4? Not clear.

- Line 215: Orthogonal Experimental Design is not referenced - is it the authors' own method?

- Are the resulting bottles real products that the study subjects could touch, are they photographs of those products, or are they illustrations like those in Figure 3?

- In lines 203 and 241, the full name is repeated with the initials of the method, when it is appropriate to only cite the full name on the first occurrence.

- The description of the methodology used to assess the perception of volunteers in the studies carried out is not sufficiently clear.

- What is the basis for the statement in line 388: The wine familiarity was also not significant influenced the results.

Reviewer 3 Report

The paper shows an interesting investigation aimed at evaluating implict and explicit differences among western and eastern people, in evaluating product packaging.

The argument deservers particular attention, and studies like that presented by the authors are important. The scientific soundness of the approach is good, as well as the distribution of the contents.

I have some concerns about the results presented in figures 5 and 7. In both figures, Germans tend to assign lower scores. However, I'm not sure that the available data are sufficient to generalize the results. Indeed, a systematic error in the experimental procedure could be present. It is not clear if the same speaker/experimenter introduced the test to two samples (I suppose to not). If not, different epmphasis on some parameters gave from two different speakers/experimenters could have influenced the results.

Differently, it is important to observe the different slopes of the curves in some of the presented frames of the figures (e.g. label position in Figure 5).

I suggest the auhors to better explain and argument the results exposed in Figures 5 and 7, because could be of particular interest. Indeed, the tendency towards the use of disposable packaging ( e.g. https://doi.org/10.4081/jae.2020.1088  ; https://doi.org/10.1002/pts.899) leads to important reflections about the acceptability of specific shapes, and leads to non-negligible design difficulties in order to optimize the packaging manufacturing processes.

Round 2

Reviewer 1 Report

The authors have improved the paper and adequately clarified the reviewer's concerns. The only style improvement I can suggest is to insert the standard deviations not only in tabular form but also in graphical form in the figures. However, my suggestion is only a matter of style because the data shown in the table is enough to understand the data significance.